# Analytical Performance of the Factory-Calibrated Flash Glucose Monitoring System FreeStyle Libre2^TM^ in Healthy Women

**DOI:** 10.3390/s23177417

**Published:** 2023-08-25

**Authors:** Zhuoxiu Jin, Alice E. Thackray, James A. King, Kevin Deighton, Melanie J. Davies, David J. Stensel

**Affiliations:** 1National Centre for Sport and Exercise Medicine, School of Sport, Exercise and Health Sciences, Loughborough University, Loughborough LE11 3TU, UK; z.jin@lboro.ac.uk (Z.J.); a.e.thackray@lboro.ac.uk (A.E.T.); j.a.king@lboro.ac.uk (J.A.K.); 2National Institute for Health and Care Research (NIHR) Leicester Biomedical Research Centre, University Hospitals of Leicester National Health Service (NHS) Trust and the University of Leicester, Leicester LE1 5WW, UK; melanie.davies@uhl-tr.nhs.uk; 3Nuffield Health, Epsom, Surrey KT18 5AL, UK; kevin.deighton@nuffieldhealth.com; 4Diabetes Research Centre, University of Leicester, Leicester LE5 4PW, UK; 5Faculty of Sport Sciences, Waseda University, Tokorozawa 359-1192, Japan; 6Department of Sports Science and Physical Education, The Chinese University of Hong Kong, Ma Liu Shui, Hong Kong 999077, China

**Keywords:** continuous glucose monitoring, blood glucose, agreement, glycaemic dynamics

## Abstract

Continuous glucose monitoring (CGM) is used clinically and for research purposes to capture glycaemic profiles. The accuracy of CGM among healthy populations has not been widely assessed. This study assessed agreement between glucose concentrations obtained from venous plasma and from CGM (FreeStyle Libre2^TM^, Abbott Diabetes Care, Witney, UK) in healthy women. Glucose concentrations were assessed after fasting and every 15 min after a standardized breakfast over a 4-h lab period. Accuracy of CGM was determined by Bland–Altman plot, 15/15% sensor agreement analysis, Clarke error grid analysis (EGA) and mean absolute relative difference (MARD). In all, 429 valid CGM readings with paired venous plasma glucose (VPG) values were obtained from 29 healthy women. Mean CGM readings were 1.14 mmol/L (95% CI: 0.97 to 1.30 mmol/L, *p* < 0.001) higher than VPG concentrations. Ratio 95% limits of agreement were from 0.68 to 2.20, and a proportional bias (slope: 0.22) was reported. Additionally, 45% of the CGM readings were within ±0.83 mmol/L (±15 mg/dL) or ±15% of VPG, while 85.3% were within EGA Zones A + B (clinically acceptable). MARD was 27.5% (95% CI: 20.8, 34.2%), with higher MARD values in the hypoglycaemia range and when VPG concentrations were falling. The FreeStyle Libre2^TM^ CGM system tends to overestimate glucose concentrations compared to venous plasma samples in healthy women, especially during hypoglycaemia and during glycaemic swings.

## 1. Introduction

Continuous glucose monitoring (CGM) devices automatically track interstitial glucose concentrations via a small sensor attached to the upper arm or abdomen. CGM is commonly used in clinical practice to help people with diabetes maintain healthy glycaemic management and prevent hypoglycaemic episodes. Studies have demonstrated high accuracy of CGM devices in individuals with diabetes [1,2], supporting the safety of CGM systems for informing treatment decisions in adults with type 1 diabetes or individuals with type 2 diabetes who are on insulin [3,4,5].

Along with clinical applications, CGM systems are also widely used for research [6,7,8,9,10] to provide comprehensive glycaemic profiles, including glucose nadir [7], glucose peak [9] and area under the curve [10]. These devices are often utilised in studies under free-living conditions, as they remove the barrier of collecting blood samples in the laboratory. However, given the limited studies conducted in healthy individuals [11,12] and the various types of CGM systems on the market [13], we are unable to draw firm conclusions on the accuracy of specific CGM systems. A study led by Akintola and colleagues [11] identified good agreement of the Medtronic ENLiTE CGM system with venous serum glucose concentrations in normoglycaemic participants. Another study also showed overall high accuracy of the Dexcom G6 CGM system compared with capillary blood glucose concentrations in healthy individuals [12].

The FreeStyle Libre^TM^ CGM system is also commonly used for both clinical and research applications [8,14,15,16]. The FreeStyle Libre^TM^ sensors are factory-calibrated, which reduces the likelihood of inaccuracies compared to user-calibrated devices such as the Medtronic ENLiTE and Dexcom. Some studies have used the first-generation FreeStyle Libre^TM^ CGM system to calculate daily glycaemic variables in healthy adults [8,16] but its accuracy compared with venous blood measurements has not been established in this population. Since the second generation of the sensor (FreeStyle Libre2^TM^) received CE mark clearance in the EU in 2018 and FDA approval in 2020, the accuracy of FreeStyle Libre2^TM^ sensors has been tested in people with type 1 diabetes or type 2 diabetes only [17], but not in healthy individuals. At least one study has observed differences in glycaemic variables assessed by CGM between normoglycemic individuals and people with impaired glucose tolerance [18], showing higher daytime average interstitial glucose concentrations and higher postprandial peaks in a group with abnormal glucose tolerance. In addition, studies examining the accuracy of CGM systems compared with glucose measurements from venous blood have provided inconsistent findings [1,15]. Therefore, the present study compared glucose responses assessed using the FreeStyle Libre2^TM^ CGM sensor with those obtained from venous plasma samples to investigate the accuracy of the sensor and its suitability for research purposes in healthy women.

## 2. Methods

### 2.1. Participants

Thirty healthy, normal-weight females provided written informed consent to participate in this study. One participant did not have sensor data; thus, data from 29 participants were subsequently integrated into the study analysis. Eligibility criteria were female, 18 to 35 years old; body mass index between 18.5 and 24.9 kg/m^2^; non-smokers (vaping was considered smoking); no known medical conditions; not taking any medications that might influence the study outcomes including those that may influence the CGM readings during the lab study period such as acetaminophen (paracetamol), acetylsalicylic acid (aspirin) or ascorbic acid (vitamin C); habitually consumed three meals a day; no clinically diagnosed eating disorders; not dieting and stable weight for 3 months before the study (<3 kg change in weight); no severe dislike or allergy to any of the study foods; and regular menstrual cycle in the past 6 months (those taking the oral contraceptive pill were not eligible for the study). Study procedures were approved by the Loughborough University Ethics Committee and complied with the Helsinki Declaration guidelines. Participants were informed of the purpose, procedures and potential risks in the study before providing written informed consent.

### 2.2. Study Design

The study was a single-arm trial consisting of a pre-assessment visit and a main trial. On the pre-assessment visit day, anthropometric parameters were measured using standard techniques. Body mass and height were measured on an integrated scale and stadiometer in light clothes and without shoes (Seca 285, Hamburg, Germany). Waist and hip circumferences were measured with a nonelastic flexible tape while standing (Seca 201, Hamburg, Germany). A CGM sensor (FreeStyle Libre 2^TM^, Abbott Diabetes Care, Witney, UK) was then fitted. According to the manufacturer’s recommendations, the sensor was placed over the participant’s posterior upper arm. The arm on which the sensor was positioned was determined based on the participant’s preference. The pre-assessment visit was scheduled two days before the main trial. One day before the main trial, participants were required to refrain from any strenuous exercise, alcohol and caffeine. Participants were provided with the same Margherita pizza (Goodfella, Green Isle Foods Ltd., Co Kildare, Ireland, 3524 kJ, carbohydrate 90 g, protein 40 g, fat 34 g) and were instructed to consume as much of this as they desired in the evening and to avoid eating or drinking anything else except plain water from 10:00 pm. On the main trial day, participants arrived at the laboratory at 8:00 am after fasting overnight for at least 10 h. A venous cannula (Venflon 20 G/32 mm, BOC Ohmeda, Sweden) was inserted into an antecubital vein on the opposite arm to the CGM sensor. Participants rested for 30 min and then consumed a fixed breakfast within 15 min, consisting of two slices of white bread (100 g), a bowl of corn flakes (15 g), bananas (150 g, weighed with the skin), strawberry jam (15 g) and semi-skimmed milk (200 g) (2121 kJ, carbohydrate 95.2 g, protein 18.2 g, fat 5.6 g). A timer was started once participants finished the breakfast (t = 0 min). Throughout the testing period, participants remained sedentary within the laboratory, leaving them free to read, watch videos or use a computer. The study protocol is shown in Figure 1.

### 2.3. FreeStyle Libre2^TM^ CGM Sensor

The current study used a sensor-based flash glucose monitoring system (FreeStyle Libre2^TM^; Abbott Diabetes Care, Witney, UK). This factory-calibrated sensor continuously monitors interstitial glucose concentration utilising wired enzyme technology (osmium mediator and glucose oxidase enzyme co-immobilized on an electrochemical sensor). Real-time glucose concentrations were obtained by scanning the sensor every 15 min via the Librelink application on participants’ smartphones. A glucose trend arrow (indicating rate and direction of change in glucose concentration) was also displayed on the screen. The trend arrows are categorised into rising quickly (increasing > 0.111 mmol/L/min), rising (increasing 0.056–0.111 mmol/L/min), changing slowly (not increasing/decreasing > 0.056 mmol/L/min), falling (decreasing 0.056–0.111 mmol/L/min) and falling quickly (decreasing > 0.111 mmol/L/min) [19].

### 2.4. Blood Samples

Venous blood samples were collected via cannula at baseline (before breakfast), immediately after breakfast (t = 0 min) and every 15 min from t = 0 min until 240 min for venous plasma glucose (VPG) measurement. Venous blood was drawn into pre-chilled 4.9 mL K3 EDTA tubes using the Sarstedt S-Monovette^®^ system (Sarstedt, Nümbrecht, Germany). These blood samples were then centrifuged at 3500 rpm (2054× *g*) for 10 min (Heraeus Labofuge 400R, Thermo Fisher Scientific, Waltham, MA, USA). The separated plasma was stored in a −80 °C freezer (TwinGuard, Panasonic, Kadoma, Japan) until biochemical analysis. VPG concentrations were assayed using the ABX Pentra Glucose PAP CP kit by colorimetry on a Pentra C400 clinical chemistry analyser (HORIBA Medical, Montpellier, France), with an intraassay coefficient of variation of 0.5%.

### 2.5. Statistical Analyses

A linear mixed model was used to compare the mean differences between venous plasma glucose concentrations and CGM readings, and Brysbaert and Stevens’s method [20] was used to calculate the effect size. Analyses were conducted using the ‘lme4’ package for linear mixed-effects models in R (version 4.0.5). The agreement between glucose measurements derived from CGM and venous plasma samples was assessed using Bland–Altman analysis [21]. The proportional bias was calculated using linear mixed models.

We also applied the 15/15% sensor agreement analysis (the proportion of CGM readings within 15% of the reference VPG values ≥ 5.6 mmol/L or within 15 mg/dL (0.83 mmol/L) of the reference VPG values < 5.6 mmol/L) and the Clarke error grid analysis (EGA) to measure the analytical accuracy of CGM systems according to the International Organization for Standardization criteria (ISO 15197:2013) [22]. The EGA was performed using the ‘ega’ package in R (version 4.0.5). In EGA, the diagonal represents perfect agreement between the CGM and reference VPG pairs, whereas the points below and above the line indicate under- and over-estimation of the reference VPG concentrations, respectively. The data points in zone A represent the CGM values that differ by less than 20% from the VPG reference pairs or are <3.5 mmol/L, given that the reference VPG values are also <3.5 mmol/L (indicating hypoglycaemia). The current study involved a healthy population; therefore, we customised the EGA hypoglycaemia cut-off value at 3.5 mmol/L (rather than 3.9 mmol/L) as recommended for people who do not have diabetes [23]. Readings in Zone A are typically considered safe for clinical decision-making. Zone B represents CGM values that deviate from the reference VPG values by more than 20% but would lead to benign or no treatment, thus considered acceptable but necessitating closer monitoring. Values falling within zone C may lead to overcorrect acceptable VPG values, potentially causing the actual VPG values to deviate from the target range of 3.5–10 mmol/L. Data points in zone D would result in failure to detect either hypoglycaemia (<3.5 mmol/L) or hyperglycaemia (>10 mmol/L). Zone E represents readings that are physiologically implausible and beyond clinical plausibility.

Additionally, a linear mixed model was used to calculate the mean absolute relative difference (MARD) for all paired measurements to determine the accuracy of the CGM sensor. The MARD was calculated using the following formula: mean [absolute value (CGM readings-VPG concentrations)/VPG concentrations], as previously performed in other studies [2,12,24]. We further investigated the MARD in different glycaemic ranges and during different rates of change based on the VPG concentrations, as high accuracy at rapid rates of blood glucose change is critical to capture important glycaemic variabilities. The rates of change in our study were calculated according to a previously used formula [25]:$$rate=\left(VPG_{i}-VPG_{i-1}\right)/\left(T_{i}-T_{i-1}\right)$$

Above, *T*_i_ and *T*_i−1_ are the timepoints of the ith and (i−1)th venous blood sample, respectively, and *VPG*_i_ and *VPG*_i−1_ are the VPG concentrations corresponding to their related timestamps. We calculated Pearson’s correlation coefficients (r) between CGM and VPG using a multilevel approach and calculated each participant’s correlation coefficient to determine individual differences in the agreement of CGM readings and VPG levels. Data are presented as mean (standard deviation) or mean (confidence intervals), as appropriate. Statistical analysis and graphs were performed using R version 4.0.5 software. The level of significance was set at *p* < 0.05.

## 3. Results

Sensor readings from the CGM could not be recorded for one participant due to a technical issue, so results are presented for 29 individuals. The characteristics of the participants are summarised in Table 1. Our dataset comprised 429 valid CGM readings with paired VPG measurements.

Figure 2 shows meal-related changes in glucose concentrations measured by CGM and VPG. The mean CGM sensor readings were 1.14 mmol/L higher than VPG concentrations (*p* < 0.001), with the 95% confidence interval ranging from 0.97 to 1.30 mmol/L and a large effect size d = 0.83.

A Bland–Altman plot of the raw individual glucose measurements is presented in Figure 3a, with a systematic bias of 1.14 mmol/L (95% CIs: 1.01, 1.26 mmol/L) and the 95% LoA ranging from −1.44 to 3.72 mmol/L. According to previous published statistical methods [21,26], we also used a natural logarithmic transformation to mitigate the heteroscedasticity. The transformed data are presented in Figure 3b, which shows a systematic bias of 0.20 (95% CIs: 0.17, 0.23), and the 95% LoA are between −0.38 to 0.79 on a log scale. The ratio 95% limits of agreement range from 0.68 to 2.20. This means that, in 95% of cases, the CGM readings are between 0.68 and 2.20 times the VPG concentrations. The plot displayed in Figure 3b demonstrates a proportional bias with increasing mean differences between the glucose concentrations from CGM and venous plasma as the mean of the two measurements increases.

According to the 15/15% agreement analysis, 45% of CGM readings were within ±0.83 mmol/L or ±15% of reference VPG concentrations. Furthermore, 56% of CGM readings were within ±20 mg/dL (±1.11 mmol/L) or ±20% of VPG concentrations, while 74% of CGM readings were within ±30 mg/dL (±1.67 mmol/L) or ±30% of the VPG concentrations (Figure 4).

The results obtained by EGA were as follows: 48.0% in zone A, 45.7% in zone B, 6.3% in zone D and no values in zones C and E (Figure 5). All the data points that fell into zone D suggest that CGM failed to detect blood glucose values < 3.5 mmol/L. This study recruited healthy individuals only; in this case, the data that fell into zone D indicate that CGM tends to overestimate the VPG concentrations. This could result in instances of hypoglycaemia going undetected in healthy individuals monitoring their blood glucose by CGM.

The overall MARD value for the FreeStyle Libre2^TM^ CGM system was 27.5% (95% CI: 20.8, 34.2%). The MARD values across different glycaemic zones varied, with the MARD value being elevated for those in the hypoglycaemic range (<3.5 mmol/L). Larger MARD values were also apparent when the rate of change in plasma glucose concentrations was categorised as falling and falling quickly (Table 2).

There was an average correlation coefficient (r value) of 0.63 (95% CI: 0.57, 0.68) between CGM readings and VPG concentrations using the multilevel method (*p* < 0.001). The individual correlation coefficients between CGM readings and VPG concentrations were positive for 28 participants, ranging from 0.20 to 0.96, whilst a negative correlation coefficient was identified for one participant (r = −0.24) (Figure 6). The individual glycaemic patterns exhibited a time lag between the two measurements, with the glucose peak appearing earlier with VPG than CGM in 25 participants (Figure 7).

## 4. Discussion

In this group of young, healthy, female participants, postprandial glucose concentrations determined by CGM were higher than VPG concentrations. There was good agreement between CGM and VPG concentrations when participants exhibited steady glucose concentrations, but agreement was poor during hypoglycaemia (<3.5 mmol/L) and during decreases in glucose.

CGM measures glucose concentrations from subcutaneous interstitial fluid, not blood, for which values are determined by the rate of glucose diffusion from plasma into the interstitial fluid and the rate of glucose uptake by subcutaneous tissue cells [27]. Thus, factors affecting cellular metabolism, glucose delivery and capillary permeability may alter interstitial glucose concentrations. In this assessment of 29 healthy individuals, glucose concentrations obtained from CGMs were higher than those measured in venous plasma, which is in line with findings reported previously [11]. Specifically, Akintola and colleagues investigated the accuracy of the Medtronic ENLiTE CGM system in participants with normoglycaemia and reported higher concentrations of glucose obtained via CGM than those obtained from venous serum samples during the day. Conversely, this study demonstrated lower CGM readings compared with paired VPG concentrations during night-time. Glucose concentrations in our study were assessed using CGM and venous sampling from the cubital vein, rather than arterial or capillary sources. After meal ingestion, blood glucose is absorbed by the small intestine, subsequently entering the bloodstream, and then diffusing from capillaries to interstitial fluid. After this exchange, the blood, which still carries some remaining glucose, flows into the veins. Therefore, in individuals with healthy cellular function, the CGM would be expected to elicit higher values than VPG as glucose enters the interstitial fluid prior to its journey towards the veins. However, this paradigm can vary notably in individuals with diabetes. Studies recruiting people with diabetes or prediabetes report higher blood glucose concentrations derived from venous blood than those obtained from CGM devices [28,29]. For those with diabetes, insulin levels which are insufficient to facilitate the uptake of glucose into cells and/or insulin resistance may result in higher values in VPG. This suggests that there might be differences in glucose metabolism between people with and without diabetes. For healthy individuals, the overestimation of CGM may lead to an erroneous perception of having hyperglycaemia.

We used the Bland–Altman method to assess the agreement between glucose concentrations obtained from a CGM sensor and VPG samples. Our study demonstrated a proportional bias between the mean and mean difference values of CGM and VPG, consistent with previous results in healthy adults [30], suggesting a greater degree of inaccuracy at higher circulating glucose concentrations.

There are no established guidelines for evaluating the measurement performance of CGM systems. The accuracy criteria currently used for CGM systems are identical to those for self-monitoring blood glucose systems (SMBG), including statistics calculated from paired references and sensor glucose determinations [31]. The International Organization for Standardization (ISO) is a global association of national standardisation bodies, and the ISO 15197:2013 document has specified two requirements for SMBG [22]: (1) at least 95% of SMBG results must be within 0.83 mmol/L of comparators of <5.6 mmol/L or within 15% of comparator glucose values ≥ 5.6 mmol/L (15/15% agreement analysis), compared to a traceable laboratory method; (2) at least 99% of results must fall within zones A and B in an EGA. The current study found that only 45% of the total CGM readings were within ±15 mg/dL (±0.83 mmol/L) or ±15% of venous reference values, 56% were within ±20 mg/dL (±1.11 mmol/L) or ±20% of venous reference values and 74% were within ±30 mg/dL (±1.67 mmol/L) or ±30% of venous reference values. The EGA shows that 85% of the CGM readings were in zones A and B. Altogether, these data indicate suboptimal accuracy of CGM sensors compared to glucose concentrations measured from venous samples.

The MARD parameter is also commonly used to characterise the analytical performance in CGM systems. A round table conference held in 2013 relating to CGM reached a consensus that MARD values should be less than 14% and values higher than 18% represent poor accuracy [32]. Our study identified an overall 27.5% (95% CIs: 20.8, 34.2%) MARD value of the FreeStyle Libre2^TM^ sensor, indicative of poor accuracy. A pooled analysis of data from previous studies using the first generation of the FreeStyle Libre^TM^ CGM devices identified an overall median absolute relative difference value (MedARD) of 12.7% (IQR 5.9–23.5%) in individuals with type 1 diabetes [15].

When calculating the MARD values in different glucose ranges, we found that MARD values were much higher in the hypoglycaemia range (<3.5 mmol/L) compared with the euglycaemia range (3.5–10 mmol/L). Due to the limited sample size and the recruitment of healthy individuals, we did not detect any hyperglycaemic values. One study has reported lower MedARD values in hyperglycaemia zones in people with type 1 diabetes [15]. Therefore, our results combined with findings from previous work, suggest that the MARD produces more relative errors during hypoglycaemia. The present study revealed that the MARD was elevated when VPG concentrations were falling. This is consistent with the findings of a study observing higher MedARD values when sensor glucose concentrations were increasing or decreasing [25]. This indicates that CGM devices may not provide accurate glucose concentrations during rapid glucose swings such as in response to food intake.

The collective findings of our study suggest a discrepancy of glucose concentrations between CGM and VPG, especially in the hypoglycaemic zone (<3.5 mmol/L) and when blood glucose levels are falling. An explanation for this discrepancy could be the physiological differences between blood and interstitial fluid. Generally, glucose concentrations in blood and interstitial fluid are similar under steady-state conditions [25,33], which is in line with our results when participants fasted. However, in the case of swings in glycaemia (defined as a rate of change > 0.055 mmol/L/min), there is a physiological time delay, with previous work suggesting that the FreeStyle Libre2^TM^ sensor exhibits a 2-min time lag [17]. This time delay is also displayed in Figure 7, which shows that peaks in VPG were generally lower and occurred sooner than peaks identified by CGM. The higher MARD value while VPG is falling could partly be explained by the time lag.

A small sample size and high variability between individuals are limitations in our study. The data were constrained practically and financially on the number of participants that could be recruited. There are known physiological differences between the sexes in glucose homeostasis, with women tending to have higher levels of insulin sensitivity compared to men [34,35]. Therefore, we restricted our recruitment to a single sex to avoid any confounding influence of sex on the study outcomes. For all the women included in the study, we calculated each individual’s correlation coefficient, which indicates the variance in the alignment of the participant’s intrinsic physiology with the CGM calibration algorithm. The extent of this inter-individual variability depends on many factors. We standardised metabolism by discouraging vigorous physical activity and prescribing a standardised evening meal the evening before the main trial. However, we could not standardise glucose concentration in different tissues in individuals to minimise the physiological time lag during rapid changes in blood glucose, as glucose is transferred from the capillary endothelium to the interstitial fluid by simple diffusion down a concentration gradient. In addition, insertion of the sensor could cause trauma at the site, disrupting tissue structure and causing an inflammatory reaction that consumes glucose. Several participants reported discomfort when the sensor was inserted, and several participants removed the sensor immediately at the end of the trial. Moreover, the implanted glucose sensor could have been placed far from a blood vessel, which may cause an extended delay in the interchange between interstitial fluid and venous blood [36]. Future studies may be improved by employing a larger sample size of healthy participants, assessing glucose in different situations (e.g., fasted, fed, exercising) and measuring glucose over multiple days to help confirm and improve the accuracy of CGM systems.

## 5. Conclusions

Our study reports suboptimal accuracy of the FreeStyle Libre 2^TM^ CGM sensor for measuring glucose concentrations compared to values obtained from venous plasma samples in young, healthy women, especially during hypoglycaemia and during glycaemic swings. A correction may need to be applied by the manufacturer and/or researcher(s) if glucose data from the CGM are compared to other compartments such as arterial, venous or capillary glucose concentrations. For research purposes, if a study design contains a non-euglycemic range or a rapid change in blood glucose concentrations, CGM devices may not accurately capture blood glucose concentrations during such periods. Since there are irremediable differences between CGM and VPG, as they present glucose concentrations in two different compartments, it would be worthwhile to consider which one is more related to pertinent physiological activities.

## Figures and Tables

**Figure 1 sensors-23-07417-f001:**
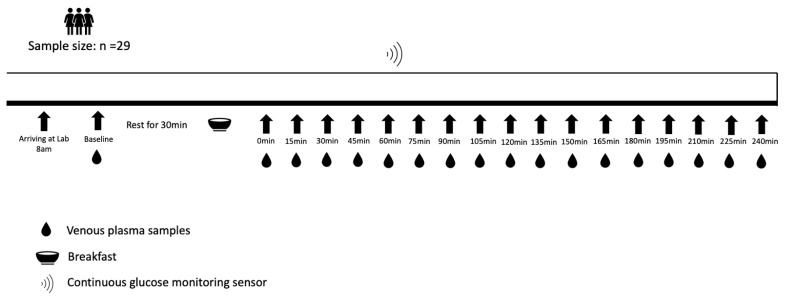
Study protocol.

**Figure 2 sensors-23-07417-f002:**
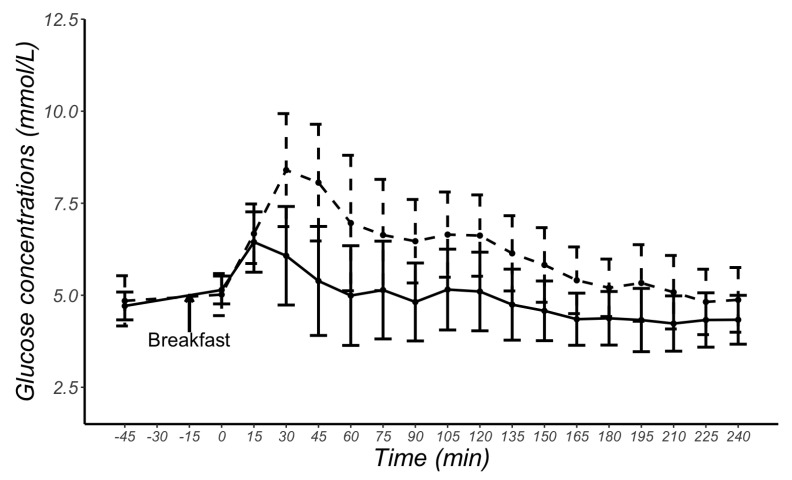
Mean and standard deviation glucose concentrations at each time point obtained from continuous glucose monitoring sensors (dashed line) and from venous plasma glucose samples (solid line) in 29 female participants.

**Figure 3 sensors-23-07417-f003:**
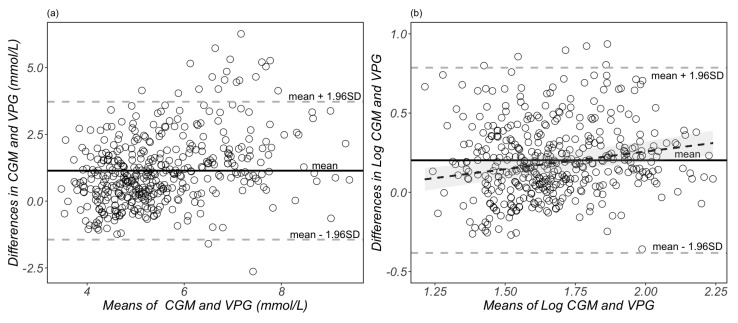
Bland–Altman plots of glucose measurements from raw (**a**) and natural log-transformed (**b**) data. Each dot represents a paired (continuous glucose monitoring and venous plasma) glucose measurement (*n* = 429 data points derived from 29 participants). The bias of the measurements and limits of agreement are represented as the solid black lines and grey dotted lines, respectively. The black dotted line and shaded grey area in (**b**) represents the proportional bias (slope = 0.22) and the 95% confidential intervals. CGM, continuous glucose monitoring; VPG, venous blood plasma glucose concentrations.

**Figure 4 sensors-23-07417-f004:**
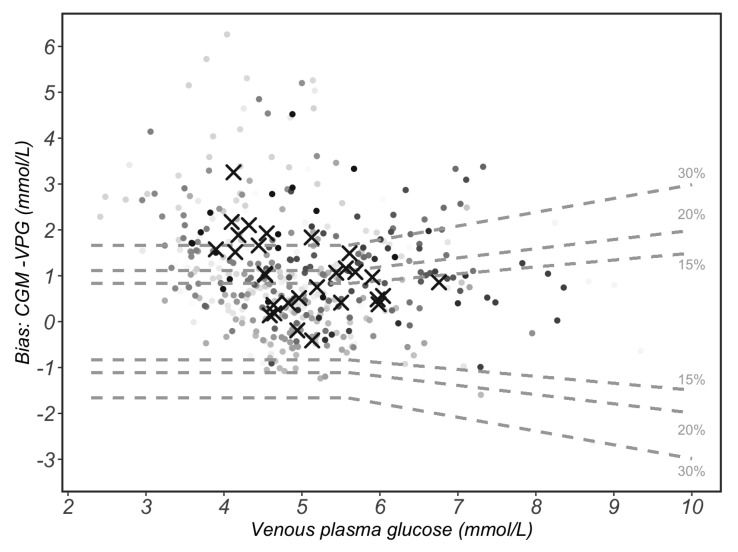
The 15/15% sensor agreement analysis. The 15% grey dotted lines form the area that continuous glucose monitoring (CGM) values differ by less than 15% of the paired and venous plasma glucose (VPG) reference values when VPG values are ≥5.6 mmol/L or differ by less than 15 mg/dL (0.83 mmol/L) of VPG values when VPG are <5.6 mmol/L. These lines also indicate the 20%/20 mg/dL (1.1 mmol/L) and 30%/30 mg/dL (1.67 mmol/L) agreement rates. Circles with varied grey shading represent each of the 429 paired measurements. Circles with same grey shade belong to the same participant. Black crosses represent the mean value of each participant (*n* = 29).

**Figure 5 sensors-23-07417-f005:**
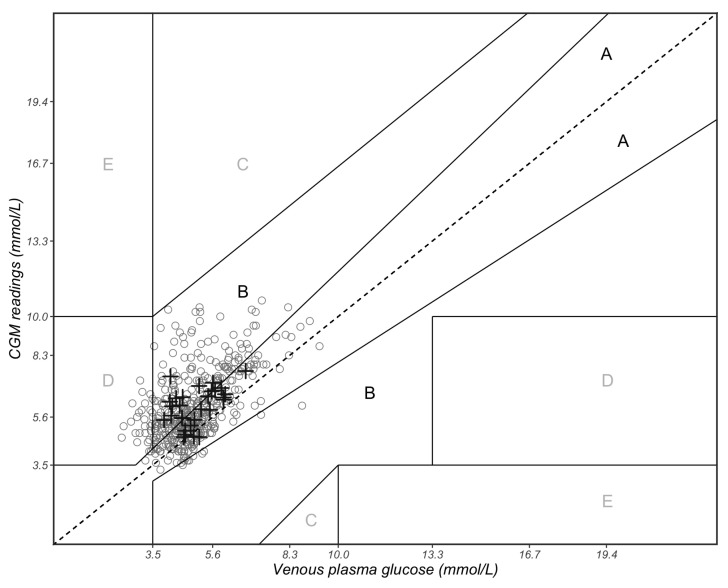
Clarke error grid analysis of paired continuous glucose monitoring (CGM) and venous plasma glucose (VPG) measurements. Zone A represents CGM values that differ by less than 20% of the VPG reference values or are <3.5 mmol/L if the reference values are also <3.5 mmol/L (indicating hypoglycaemia). Zone B represents CGM values that deviate from the reference VPG concentrations by >20% but would result in benign or no treatment. Zones A and B are clinically acceptable, while values in zones C, D and E are potentially dangerous and, therefore, clinically significant. The solid circles represent the 429 paired measurements, and the black crosses represent the mean value of each of the 29 participants.

**Figure 6 sensors-23-07417-f006:**
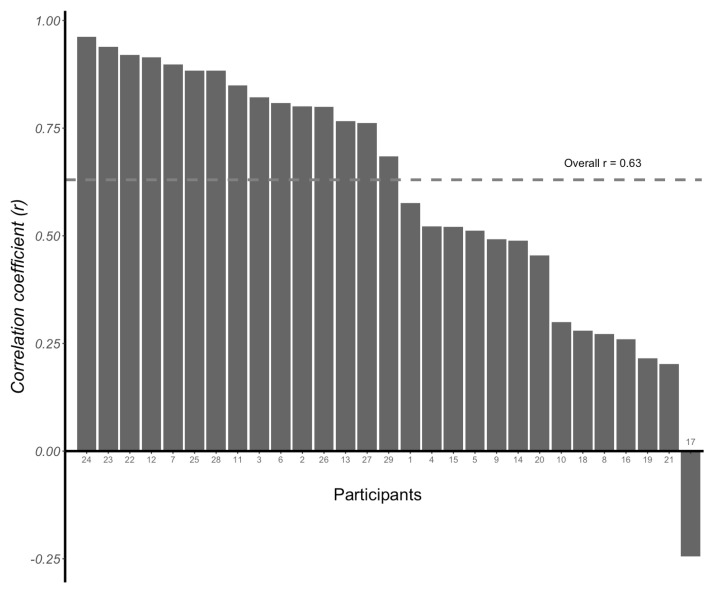
Individual correlation coefficients (r values) between venous plasma glucose concentrations and continuous glucose monitoring (CGM) readings determined for each of the 29 participants. Overall correlation coefficient r = 0.63 (95% CI: 0.57, 0.68).

**Figure 7 sensors-23-07417-f007:**
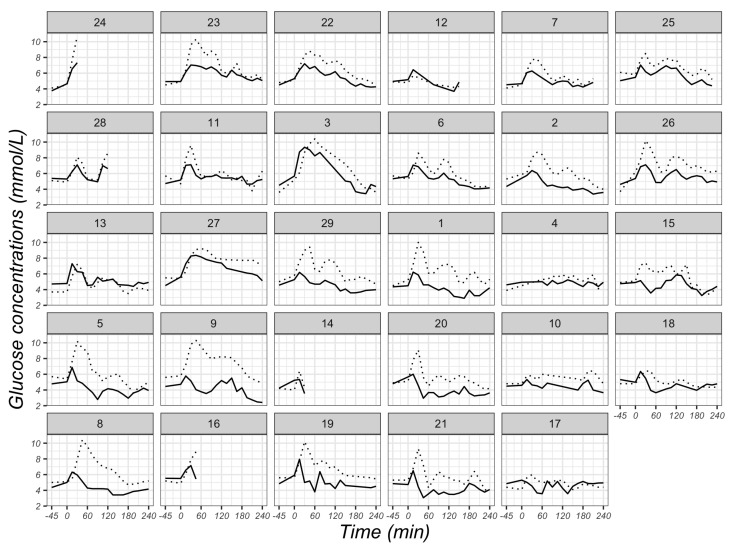
Individual glucose concentrations at different time points derived from venous plasma (solid lines) and CGM sensors (dotted lines) in 29 participants. Glucose concentrations were measured fasted at baseline (−45 min), immediately after breakfast consumption (0 min) and at 15-min intervals thereafter in the postprandial state until 240 min after breakfast consumption. The number of the individual plot represents the ID of each participant, and the order of plots is dependent on the magnitude of the r value from each participant in Figure 6.

**Table 1 sensors-23-07417-t001:** Characteristics of participants (*n* = 29 women).

	Mean ± SD
Age (years)	26 ± 4
Height (m)	1.66 ± 0.07
Body mass (kg)	60.0 ± 7.2
Body mass index (kg/m^2^)	21.5 ± 2.0
Waist circumference (cm)	69.9 ± 5.3
Hip circumference (cm)	95.7 ± 4.1
Ethnicity	
Asian	22
White	4
Arab	2
Latino	1

**Table 2 sensors-23-07417-t002:** Mean absolute relative difference (MARD) for 429 paired CGM and venous plasma glucose measurements obtained in 29 women.

	MARD (95% CI)
Overall (*n* = 429)	27.5% (20.8, 34.2%)
Glycaemic zones	
Hypoglycaemia (<3.5 mmol/L; *n* = 29)	69.5% (51.4, 82.9%)
Euglycaemia (3.5–10.0 mmol/L; *n* = 400)	25.9% (19.9, 31.8%)
Rate of changes	
Rising quickly (>0.111 mmol/L/min; *n* = 12)	14.2% (9.6, 18.8%)
Rising (0.056–0.111 mmol/L/min; *n* = 34)	13.8% (9.9, 17.8%)
Changing slowly (<0.056 mmol/L/min; *n* = 313)	28.4% (21.2, 35.6%)
Falling (0.056–0.111 mmol/L/min; *n* = 26)	63.7% (45.7, 81.7%)
Falling quickly (>0.111 mmol/L/min; *n* = 8)	67.3% (33.9, 100%)

Abbreviations: MARD, mean absolute relative difference; CIs, confidence intervals.

## Data Availability

The data presented in this study are accessible through the Loughborough University Research Repository. https://repository.lboro.ac.uk/articles/dataset/Continuous_glucose_monitor_CGM_and_venous_plasma_glucose_VPG_data_for_Exploration_of_the_correlation_between_glucose_dynamics_and_energy_intake_in_women/23814873.

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
