# Peer review of "Analytical Performance of the Factory-Calibrated Flash Glucose Monitoring System FreeStyle Libre2TM in Healthy Women"

_sensors, 2023, doi:10.3390/s23177417_

Round 1

Reviewer 1 Report

The work by Jin z et al., titled "Analytical Performance of the Factory-Calibrated Flash Glucose Monitoring System Freestyle Libre2 in Healthy Women", is very well designed and analysed, in addition to providing new knowledge, it only requires that the authors make greater precisions, such as the size of the sample (L277).

L59.- Authors should indicate if subjects were not taking medications such as paracetamol or ascorbic acid, which may interfere with similar devices such as the Dexcom G7.

L205.- Without taking into account the rapid rise in insulin, why did 29 of them have hypoglycaemia, given that these are healthy subjects with a body mass index (kg/m2) of 21.5 ± 2.0, was there any relationship with the type of breakfast?

L231.- The question seems obvious. Why compare glucose from subcutaneous interstitial fluid with VPG if they are different analytes? Wouldn't it be better to standardise and establish reference values for CGM in different populations?

Reviewer 2 Report

This is a study comparing glucose measurements made by a commercial sensor, Freestyle Libre 2TM CGM, with venous plasma measurements for a group of healthy female individuals. The authors conclude there are large differences between the two methods of measurement, questioning the accuracy of the commercial sensors. This study may be of interest to those using such sensors as patients or those in related businesses.

The study is very brief and there are some questions that need to be answered, in addition to some minor presentation/writing issues:

Line 4: "and from" rather than "versus" in this construction

Line 5: either "over 4 hours" or "over a 4-hour period". Also, when it's a number less than ten, it is customary to present it as a numeral written in full, i.e. "over four hours"

Line 9: I do not understand the "22" before "mmol/L"

Line 42: received (cross out "has")

Study group is too small, only 29 people.

Also, why only females? Why have you not included males also?

You need to include a proper experimental section on chemicals/materials used. Solvents, EDTA, etc. from what vendor? With what purity?

Line 108: Mention brand/type of centrifuge. Mention g force.

Line 109: Mention brand/type of freezer.

Lines 207-209: My logic tells me that such factors would lead to a delay (or "loss on the route") of glucose, so I would expect lower, not higher glucose readings on the CGV versus VPG. How do you (or the literature, since you mention precedence with other brands of similar sensors) explain the consistently positive error of sensor measurements?

Line 218: glucose metabolism kinetics (you need a process, a substance/glucose does not have kinetics, a process has associated kinetics)

The systematically positive error observed suggests that there may be an incorrect calibration of the CGM sensors. Please add information about how the initial calibration of the sensors was performed.

The conclusion is weak. You should provide advice to the manufacturers of such sensors, for example an equation or some other sort of correction that needs to be applied for recalibrating the sensors such that their readings approach real values better.

References: DOIs are missing. Please add them.  

English is fine. Only minor glitches here and there.
